# Hamiltonian Mechanics of Feature Learning: Bottleneck Structure in Leaky ResNets

Arthur Jacot[1], Alexandre Kaiser[1]
[1]Courant Institute, NYU
arthur.jacot@nyu.edu,amk1004@nyu.edu

We study Leaky ResNets, which interpolate between ResNets and Fully-Connected nets depending on an 'effective depth' hyper-parameter $\tilde{L}$. In the infinite depth limit, we study 'representation geodesics' $A_p$: continuous paths in representation space (similar to NeuralODEs) from input $p = 0$ to output $p = 1$ that minimize the parameter norm of the network. We give a Lagrangian and Hamiltonian reformulation, which highlight the importance of two terms: a kinetic energy which favors small layer derivatives $\partial_p A_p$ and a potential energy that favors low-dimensional representations, as measured by the 'Cost of Identity'. The balance between these two forces offers an intuitive understanding of feature learning in ResNets. We leverage this intuition to explain the emergence of a bottleneck structure, as observed in previous work: for large $\tilde{L}$ the potential energy dominates and leads to a separation of timescales, where the representation jumps rapidly from the high dimensional inputs to a low-dimensional representation, move slowly inside the space of low-dimensional representations, before jumping back to the potentially high-dimensional outputs. Inspired by this phenomenon, we train with an adaptive layer step-size to adapt to the separation of timescales.

## 1. Introduction

Feature learning is generally considered to be at the center of the recent successes of deep neural networks (DNNs), but it also remains one of the least understood aspects of DNN training.

There is a rich history of empirical analysis of feature learning, for example the appearance of local edge detections in CNNs with a striking similarity to the biological visual cortex [1], feature arithmetic properties of word embeddings [2], similarities between representations at different layers [3, 4], or properties such as Neural Collapse [5] to name a few. While some of these phenomena have been studied theoretically [6–8], a general theory of feature learning in DNNs is still lacking.

For shallow networks, there is now strong evidence that the first weight matrix is able to recognize a low-dimensional projection of the inputs that determines the output (assuming this structure is present) [9–11]. A similar phenomenon appears in linear networks, where the network is biased towards learning low-rank functions and low-dimensional representations in its hidden layers [12–14]. But in both cases the learned features are restricted to depend linearly on the inputs, and the feature learning happens in the very first weight matrix, whereas it has been observed that features increase in complexity throughout the layers [15].

The linear feature learning ability of shallow networks has inspired a line of work that postulates that the weight matrices learn to align themselves with the backward gradients and that by optimizing for this alignment directly, one can achieve similar feature learning abilities even in deep nets [16, 17].

For deep nonlinear networks, a theory that has garnered a lot of interest is the Information Bottleneck [18], which observed amongst other things that the inner representations appear to maximize their mutual information with the outputs, while minimizing the mutual information with the inputs. A limitation of this theory is its reliance on the notion of mutual information which has no obvious definition for empirical distributions, which led to some criticism [19].

Second Conference on Parsimony and Learning (CPAL 2025).

A recent theory that is similar to the Information Bottleneck but with a focus on the dimensionality/rank of the representations and weight matrices rather than the mutual information is the Bottleneck rank/Bottleneck structure [20–22]: which describes how, for large depths, most of the representations will have approximately the same low dimension, which equals the Bottleneck rank of the task (the minimal dimension that the inputs can be projected to while still allowing for fitting the outputs). The intuitive explanation for this bias is that a smaller parameter norm is required to (approximately) represent the identity on low-dimensional representations rather than high dimensional ones. Some other types of low-rank bias have been observed [23, 24].

In this paper we will focus on describing the Bottleneck structure in ResNets, and formalize the notion of 'cost of identity' as a driving force for the bias towards low dimensional representation. The ResNet setup allows us to consider the continuous paths in representation space from input to output, similar to the NeuralODE [25], and by adding weight decay, we can analyze representation geodesics, which are paths that minimize parameter norm, as already studied in [26]. The appearance of separation of timescales in the layers of ResNets with a modified loss has been mentioned in [27], under the name 'turnpike principle', but the underlying mechanism for the separation of timescales/turnpike behavior are very different to ours and no low-dimensional bias is observed.

## 2. Leaky ResNets

Our goal is to study a variant of the NeuralODE [25, 26] approximation of ResNet with leaky skip connections and with $L_2$-regularization. The classical NeuralODE describes the continuous evolution of the activations $\alpha_p(x) \in \mathbb{R}^w$ starting from $\alpha_0(x) = x$ at the input layer $p = 0$ and then follows

$$\partial_p \alpha_p(x) = W_p \sigma(\alpha_p(x))$$

for the $w \times (w + 1)$ matrices $W_p$ and the nonlinearity $\sigma : \mathbb{R}^w \to \mathbb{R}^{w+1}$ which maps a vector $z$ to $\sigma(z) = ([z_1]_+, \ldots, [z_w]_+, 1)$, applying the ReLU nonlinearity entrywise and appending a new entry with value 1. Thanks to the appended 1 we do not need any explicit bias, since the last column $W_{p, \cdot w+1}$ of the weights replaces the bias.

This can be thought of as a continuous version of the traditional ResNet with activations $\alpha_\ell(x)$ for $\ell = 1, \ldots, L$: $\alpha_{\ell+1}(x) = \alpha_\ell(x) + W_\ell \sigma(\alpha_\ell(x))$.

We will focus on **Leaky ResNets**, a variant of ResNets that interpolate between ResNets and Fully-Connected Neural Networks (FCNNs), by tuning the strength of the skip connections leading to the following ODE with parameter $\tilde{L}$:

$$\partial_p \alpha_p(x) = -\tilde{L}\alpha_p(x) + W_p \sigma(\alpha_p(x)).$$

This can be thought of as the continuous version of $\alpha_{\ell+1}(x) = (1 - \tilde{L})\alpha_\ell(x) + W_\ell \sigma(\alpha_\ell(x))$. As we will see, the parameter $\tilde{L}$ plays a similar role as the depth in a FCNN.

Finally we will be interested in describing the paths that minimize a cost with $L_2$-regularization

$$\min_{W_p} \frac{1}{N} \sum_{i=1}^N \|f^*(x_i) - \alpha_1(x_i)\|^2 + \frac{\lambda}{2\tilde{L}} \int_0^1 \|W_p\|_F^2 \, dp.$$

The scaling of $\frac{\lambda}{\tilde{L}}$ for the regularization term will be motivated in Section 2.1.

This type of optimization has been studied in [26] without leaky connections. In this paper, we describe the large $\tilde{L}$ behavior which leads to a so-called Bottleneck structure [20, 21] as a result of a separation of timescales in $p$.

### 2.1. A Few Symmetries

Changing the leakage parameter $\tilde{L}$ is equivalent (up to constants) to changing the integration range $[0, 1]$ or to scaling the outputs.

**Integration range:** Consider the weights $W_p$ on the range $[0, 1]$ and leakage parameter $\tilde{L}$, leading to activations $\alpha_p$. Then stretching the weights to a new range $[0, c]$, by defining $W'_q = \frac{1}{c} W_{q/c}$ for $q \in [0, c]$, and dividing the leakage parameter by $c$, stretches the activations $\alpha'_q = \alpha_{q/c}$:

$$\partial_q \alpha'_q(x) = -\frac{\tilde{L}}{c} \alpha'_q(x) + \frac{1}{c} W_{q/c} \sigma(\alpha'_q(x)) = \frac{1}{c} \partial_p \alpha_{q/c}(x),$$

and the parameter norm is simply divided by $c$: $\int_0^c \|W'_q\|^2 \, dq = \frac{1}{c} \int_0^1 \|W_p\|^2 \, dp$.

This implies that a path on the range $[0, c]$ with leakage parameter $\tilde{L} = 1$ is equivalent to a path on the range $[0, 1]$ with leakage parameter $\tilde{L} = c$ up to a factor of $c$ in front of the parameter weights. For this reason, instead of modeling different depths as changing the integration range, we will keep the integration range to $[0, 1]$ for convenience but change the leakage parameter $\tilde{L}$ instead. To get rid of the factor in front of the integral, we choose a regularization term of the form $\frac{\lambda}{\tilde{L}}$. From now on, we call $\tilde{L}$ the (effective) depth of the network.

Note that this also suggests that in the absence of leakage ($\tilde{L} = 0$), changing the range of integration has no effect on the effective depth, since $2\tilde{L} = 0$ too. Instead, in the absence of leakage, the effective depth can be increased by scaling the outputs as we now show.

**Output scaling:** Given a path $W_p$ on the $[0, 1]$ (for simplicity, we assume that there are no bias, i.e. $W_{p, \cdot w+1} = 0$), then increasing the leakage by a constant $\tilde{L} \to \tilde{L} + c$ leads to a scaled down path $\alpha'_p = e^{-cp} \alpha_p$. Indeed we have $\alpha'_0(x) = \alpha_0(x)$ and

$$\partial_p \alpha'_p(x) = -(\tilde{L} + c)\alpha'_p(x) + W_p \sigma(\alpha'_p(x)) = e^{-cp} \left( \partial_p \alpha_p(x) - c\alpha_p(x) \right) = \partial_p(e^{-cp} \alpha_p(x)).$$

Thus a nonleaky ResNet $\tilde{L} = 0$ with very large outputs $\alpha_1(x)$ is equivalent to a leaky ResNet $\tilde{L} > 0$ with scaled down outputs $e^{-\tilde{L}} \alpha_1(x)$. Such large outputs are common when training on cross-entropy loss, and other similar losses that are only minimized at infinitely large outputs. When trained on such losses, it has been shown that the outputs of neural nets will keep on growing during training [28, 29], suggesting that when training ResNets on such a loss, the effective depth increases during training (though quite slowly).

## 2.2. Lagrangian Reformulation

The optimization of Leaky ResNets can be reformulated, leading to a Lagrangian form.

First observe that the weights $W_p$ at any minimizer can be expressed in terms of the matrix of activations $A_p = \alpha_p(X) \in \mathbb{R}^{w \times N}$ over the whole training set $X \in \mathbb{R}^{w \times N}$ (similar to [30]):

$$W_p = (\tilde{L} A_p + \partial_p A_p) \sigma(A_p)^+$$

where $(\cdot)^+$ is the pseudo-inverse. This formula comes from the fact that $W_p$ has minimal parameter norm amongst the weights $W$ that satisfy $\partial_p A_p = -\tilde{L} A_p + W \sigma(A_p)$.

We therefore consider the equivalent optimization over the activations $A_p$:

$$\min_{A_p : A_0 = X} \frac{1}{N} \|f^*(X) - A_1\|^2 + \frac{\lambda}{2\tilde{L}} \int_0^1 \left\| \tilde{L} A_p + \partial_p A_p \right\|_{K_p}^2 \, dp,$$

where the norm $\|M\|_{K_p} = \|M\sigma(A_p)^+\|_F$ corresponds to the scalar product $\langle A, B \rangle_{K_p} = \mathrm{Tr}\left[ A K_p^+ B^T \right]$ for the $N \times N$ matrix $K_p = \sigma(A_p)^T \sigma(A_p)$. By convention, we say that $\|M\|_{K_p} = \infty$ if $M$ does not lie in the image of $K_p$, i.e. $\mathrm{Im} M^T \not\subseteq \mathrm{Im} K_p$.

It can be helpful to decompose this loss along the different neurons

$$\min_{A_p : A_0 = X} \sum_{i=1}^{w} \frac{1}{N} \|f_i^*(X) - A_{1,i}\|^2 + \frac{\lambda}{2\tilde{L}} \int_0^1 \left\| \tilde{L} A_{p,i\cdot} + \partial_p A_{p,i\cdot} \right\|_{K_p}^2 \, dp,$$

Leading to a particle flow behavior, where the neurons $A_{p,i\cdot} \in \mathbb{R}^N$ are the particles. At first glance, it appears that there is no interaction between the particles, but remember that the norm $\|\cdot\|_{K_p}$ depends on the covariance $K_p = \sum_{i=1}^w \sigma((A_p)_{i\cdot})\sigma((A_p)_{i\cdot})^T$, leading to global interactions between neurons. If we assume that $\mathrm{Im}A_p^T \subset \mathrm{Im}\sigma(A_p)^T$, we can decompose the inside of the integral as three terms:

$$\frac{1}{2\tilde{L}}\left\|\tilde{L}A_p + \partial_p A_p\right\|_{K_p}^2 = \frac{\tilde{L}}{2}\|A_p\|_{K_p}^2 + \langle\partial_p A_p, A_p\rangle_{K_p} + \frac{1}{2\tilde{L}}\|\partial_p A_p\|_{K_p}^2.$$

**Cost of identity** $\|A_p\|_{K_p}^2$ / **potential energy** $-\frac{\tilde{L}}{2}\|A_p\|_{K_p}^2$: This term can be interpreted as a form of potential energy, since it only depends on the representation $A_p$ and not its derivative $\partial_p A_p$. We call it the cost of identity (COI), since it is the Frobenius norm of the smallest weight matrix $W_p$ such that $W_p\sigma(A_p) = A_p$. The COI can be interpreted as measuring the dimensionality of the representation, inspired by the fact if the representations $A_p$ is non-negative (and there is no bias $\beta = 0$), then $A_p = \sigma(A_p)$ and the COI simply equals the rank $\|A_p\|_{K_p}^2 = \mathrm{Rank}A_p$ (this interpretation is further justified in Section 2.3). We follow the convention of defining the potential energy as the negative of the term that appears in the Lagrangian, so that the Hamiltonian equals the sum of energies.

**Middle term**: The middle term $\langle\partial_p A_p, A_p\rangle_{K_p}$ ends up playing a very minor role in the analysis of this paper. This is probably because this term is invariant under reparametrizing the paths $A_{p(q)}$ for any increasing function $p : [0, 1] \to [0, 1]$, whereas the other two other terms are not. Simply speaking, it depends on the trajectory taken, but not the speed at which one traverses this trajectory. Since the main goal of this paper is to prove a separation of timescales by showing that the network moves rapidly in high-dimensional layers and slowly in low-dimensional ones, the middle term is mostly irrelevant to our analysis.

**Kinetic energy** $\frac{1}{2\tilde{L}}\|\partial_p A_p\|_{K_p}^2$: This term measures the size of the representation derivative $\partial_p A_p$ w.r.t. the $K_p$ norm. It favors paths $p \mapsto A_p$ that do not move too fast, especially along directions where $\sigma(A_p)$ is small. This interpretation as a kinetic energy also illustrates how the inverse kernel $K_p^+$ is the analogue of the mass matrix from classical mechanics.

This suggests that the local optimal paths must balance two objectives that are sometimes opposed: the kinetic energy favors going from input representation to output representation in a 'straight line' that minimizes the path length, the COI on the other hand favors paths that spends most of the path in low-dimensional representations that have a low COI. The balance between these two goals shifts as the depth $\tilde{L}$ grows, and for large depths it becomes optimal for the network to rapidly move to a representation of smallest possible dimension (but not too small that it becomes impossible to map back to the outputs), remain inside the space of low-dimensional representations for most of the layers, and finally move rapidly to the output representation; even if this means doing a large 'detour' and having a large kinetic energy. The main goal of this paper is to describe this general behavior.

Note that one could imagine that as $\tilde{L} \to \infty$ it would always be optimal to first go to the minimal COI representation which is the zero representation $A_p = 0$, but once the network reaches a zero representation, it can only learn constant representations afterwards (the matrix $K_p = \mathbf{1}\mathbf{1}^T$ is then rank 1 and its image is the space of constant vectors). So the network must find a representation that minimizes the COI under the condition that there is a path from this representation to the outputs.

*Remark.* While this interpretation and decomposition is a pleasant and helpful intuition, it is rather difficult to leverage for theoretical proofs directly. The problem is that we will focus on regimes where the representations $A_p$ and $\sigma(A_p)$ are approximately low-dimensional (since those are the representations that locally minimize the COI), leading to an unbounded pseudo-inverse $\sigma(A_p)^+$. This is balanced by the fact that $(\tilde{L}A_p + \partial_p A_p)$ is small along the directions where $\sigma(A_p)^+$ explodes, ensuring a finite weight matrix norm $\left\|\tilde{L}A_p + \partial_p A_p\right\|_{K_p}^2$. But the suppression of $(\tilde{L}A_p + \partial_p A_p)$ along these bad directions usually comes from cancellations, i.e. $\partial_p A_p \approx -\tilde{L}A_p$. In such cases, the decomposition in three terms of the Lagrangian is ill adapted since all three terms are infinite and

cancel each other to yield a finite sum $\left\|\tilde{L}A_p + \partial_p A_p\right\|^2_{K_p}$. One of our goal is to save this intuition and prove a similar decomposition with stable equivalent to the cost of identity and kinetic energy where $K_p^+$ is replaced by the bounded $(K_p + \gamma I)^+$ for the right choice of $\gamma$.

## 2.3. Cost of Identity as a Measure of Dimensionality

This section shows the relation between the COI of and the dimensionality of the data. The intuition is simple, the cost of representing the identity on a $k$-dimensional representation should be $k$, at least for representations that locally minimize the COI. These local minima of the COI are of interest because the representations inside the bottleneck are close to local minima of the COI.

We define two types of COI, the standard COI (or COI with bias) $\|A\|^2_K$ which is the one that appears in the previous sections, and the COI without bias $\|A\|^2_{\bar{K}}$, where for any activation matrix $A$, we define the covariance with bias $K = \sigma(A)^T\sigma(A)$ and without bias $\bar{K} = \bar{\sigma}(A)^T\bar{\sigma}(A)$ where $\bar{\sigma}$ denotes the simple ReLU (without appending a constant entry), leading to the relation $\bar{K} = K - 1_N 1_N^T$.

It is easier to see the relation between the COI without bias and the dimensionality of the representation. For example if the representation is nonnegative $A \geq 0$, we have $\|A\|^2_{\bar{K}} = \|A\bar{\sigma}(A)^+\|^2_F = \|AA^+\|^2_F = \operatorname{Rank}A$. More generally, the COI without bias is lower bounded by a notion of effective dimension:

**Proposition 1.** $\|A\|^2_{\bar{K}} \geq \frac{\|A\|^2_*}{\|A\|^2_F}$ for the nuclear norm $\|A\|_* = \sum_{i=1}^{\operatorname{Rank}A} s_i(A)$.

*Proof.* Since $\|\bar{\sigma}(A)\|_F \leq \|A\|_F$, we have $\|A\|^2_{\bar{K}} = \|A\bar{\sigma}(A)^+\|^2_F \geq \min_{\|B\|_F \leq \|A\|_F} \|AB^+\|^2_F$ which is minimized at $B = \frac{\|A\|_F}{\sqrt{\|A\|_*}}\sqrt{A}$ leading to a lower bound of $\|AB^+\|^2_F = \frac{\|A\|_*}{\|A\|^2_F}\left\|\sqrt{A}\right\|^2_F = \frac{\|A\|^2_*}{\|A\|^2_F}$. □

The stable rank $\frac{\|A\|^2_*}{\|A\|^2_F}$ is upper bounded by $\operatorname{Rank}A$, with equality if all non-zero singular values of $A$ are equal, and it is lower bounded by the more common notion of stable rank $\frac{\|A\|^2_F}{\|A\|^2_{op}}$, because $\sum s_i \max s_i \geq \sum s_i^2$ for the singular values $s_i$.

Note that in contrast to the COI which is a very unstable quantity because of the pseudo-inverse, the ratio $\frac{\|A\|^2_*}{\|A\|^2_F}$ is continuous except at $A = 0$. This also makes it much easier to compute empirically than the COI itself.

The relation between the COI with bias and dimensionality. is less obvious in general, but as we will see, inside the bottleneck the representation will approach local minima of the COI with bias. It turns out that at any local minima $A$ that is in some sense stable under adding more neurons, not only is the representation nonnegative, but both COIs must also match and be equal to the dimension:

**Proposition 2.** *A local minimum of $A \mapsto \|A\|^2_K$ is said to be stable if it remains a local minimum after concatenating a zero vector $A' = \begin{pmatrix} A \\ 0 \end{pmatrix} \in \mathbb{R}^{(w+1)\times N}$. All stable minima are non-negative, and satisfy $\|A\|^2_K = \|A\|^2_{\bar{K}} = \operatorname{Rank}A$.*

These stable minima will play a significant role in the rest of our analysis, as we will see that for large $\tilde{L}$ the representations $A_p$ of most layers will be close to one such local minimum. Now we are not able to rule out the existence of non-stable local minima (nor guarantee that they are avoided with high probability), but one can show that all strict local minima of wide enough networks are stable. Actually we can show something stronger, starting from any non-stable local minimum there is a constant loss path that connects it to a saddle:

**Proposition 3.** *If $w > N(N+1)$ then if $\hat{A} \in \mathbb{R}^{w\times N}$ is local minimum of $A \mapsto \|A\|^2_K$ that is not non-negative, then there is a continuous path $A_t$ of constant COI such that $A_0 = \hat{A}$ and $A_1$ is a saddle.*

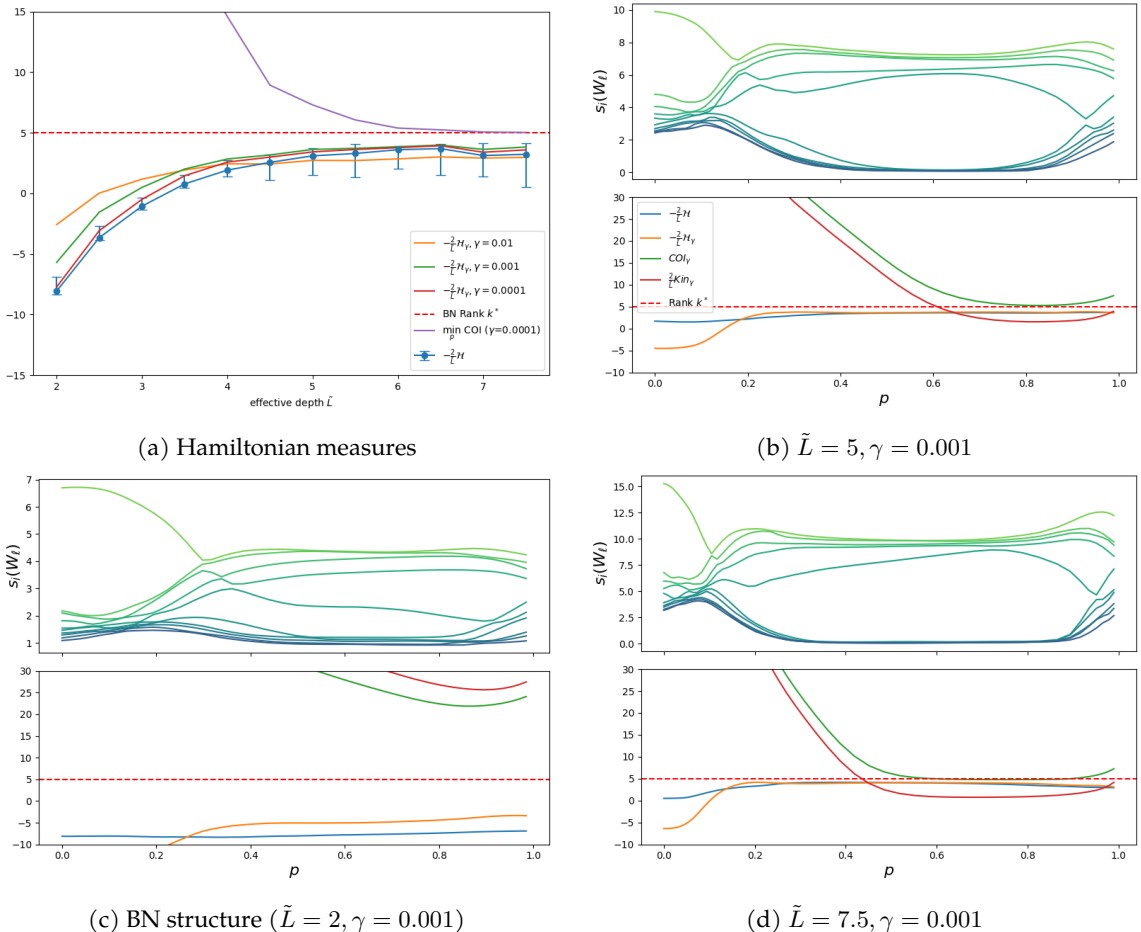

(a) Hamiltonian measures

(b) $\tilde{L} = 5, \gamma = 0.001$

(c) BN structure ($\tilde{L} = 2, \gamma = 0.001$)

(d) $\tilde{L} = 7.5, \gamma = 0.001$

Figure 1: **Leaky ResNet structures:** We train adaptive networks with a fixed $L = 50$ over a range of effective depths $\tilde{L}$. The true function $f^* : \mathbb{R}^{20} \to \mathbb{R}^{20}$ is the composition of two random FCNNs $g_1, g_2$ mapping from dim. 20 to 5 to 20, the network recovers the true rank of $k^* = 5$. (a) Estimates of the Hamiltonian constants for networks trained with different $\tilde{L}$. The Hamiltonian refers to $-\frac{2}{\tilde{L}}\mathcal{H}$ which estimates the true rank $k^*$. The COI refers to $\min_p ||A_p||_{K_p}$. The trend line follows the median estimate for $-\frac{2}{\tilde{L}}\mathcal{H}$ across each network's layers, whereas the error bars signify its minimum and maximum over $p \in [0, 1]$. The "stable" Hamiltonians utilize the relaxation from Theorem 4. (b,c,d) Top: The 10 largest singular values of $W_p$ throughout the layers, exhibiting a BN structure. Bottom: the rescaled Hamiltonian, stable Hamiltonian, COI and kinetic energy. The Hamiltonian remains constant throughout the layers, and the stable Hamiltonian approximates it well - except in the first layers, where the kinetic energy (and COI) blows up which is in line with the bound of Equation 3 in Theorem 4. Inside the bottleneck, the kinetic energy approaches zero and the COI approaches $k^*$.

This could explain why a noisy GD would avoid such negative/non-stable minima, since there is no 'barrier' between the minima and a lower one, one could diffuse along the path described in Proposition 3 until reaching a saddle and going towards a lower COI minima. But there seems to be something else that pushes away from such non-negative minima, as in our experiments with full population GD we have only observed stable/non-negative local minimas.

## 2.4. Hamiltonian Reformulation

We can further reformulate the evolution of the optimal representations $A_p$ in terms of a Hamiltonian, similar to Pontryagin's maximum principle.

Let us define the backward pass variables $B_p = -\frac{1}{\lambda}\partial_{A_p}C(A_1)$ for the cost $C(A) = \frac{1}{N}\|f^*(X) - A\|_F^2$, which play the role of the 'momenta' of $A_p$ in this Hamiltonian interpretation, and follow the backward differential equation

$$B_1 = -\frac{1}{\lambda}\partial_{A_1}C(A_1) = \frac{2}{\lambda N}(f^*(X) - A_1)$$

$$-\partial_p B_p = \dot{\sigma}(A_p) \odot \left[W_p^T B_p\right] - \tilde{L}B_p.$$

Now at any critical point, we have that $\partial_{W_p}C(A_1) + \frac{\lambda}{\tilde{L}}W_p = 0$ and thus $W_p = -\frac{\tilde{L}}{\lambda}\partial_{A_p}C(A_1)\sigma(A_p)^T = \tilde{L}B_p\sigma(A_p)^T$, leading to joint dynamics for $A_p$ and $B_p$:

$$\partial_p A_p = \tilde{L}(B_p\sigma(A_p)^T\sigma(A_p) - A_p)$$

$$-\partial_p B_p = \tilde{L}\left(\dot{\sigma}(A_p) \odot \left[\sigma(A_p)B_p^T B_p\right] - B_p\right).$$

These are Hamiltonian dynamics $\partial_p A_p = \partial_{B_p}\mathcal{H}$ and $-\partial_p B_p = \partial_{A_p}\mathcal{H}$ w.r.t. the Hamiltonian

$$\mathcal{H}(A_p, B_p) = \frac{\tilde{L}}{2}\left\|B_p\sigma(A_p)^T\right\|^2 - \tilde{L}\mathrm{Tr}\left[B_p A_p^T\right]. \tag{1}$$

The Hamiltonian is a conserved quantity, i.e. it is constant in $p$. It will play a significant role in describing a separation of timescales that appears for large depths $\tilde{L}$. Another significant advantage of the Hamiltonian reformulation over the Lagrangian approach is the absence of the unstable pseudo-inverses $\sigma(A_p)^+$.

*Remark.* Note that the Lagrangian and Hamiltonian reformulations have already appeared in previous work [26] for non-leaky ResNets. Our main contributions are the description in the next section of the Hamiltonian as the network becomes leakier $\tilde{L} \to \infty$, the connection to the cost of identity, and the appearance of a separation of timescales. These structures are harder to observe in non-leaky ResNets (though they could in theory still appear since increasing the scale of the outputs is equivalent to increasing the effective depth $\tilde{L}$ as shown in Section 2.1).

The Lagrangian and Hamiltonian are also very similar to the ones in [31, 32], and the separation of timescales and rapid jumps that we will describe also bear a strong similarity. Though a difference with our work is that the norm $\|\cdot\|_{K_p}$ depends on $A_p$ and can be degenerate.

## 3. Bottleneck Structure in Representation Geodesics

In deep fully-connected networks, a so-called Bottleneck structure emerges [20, 21], where the weight matrices and representations in the middle layers are approximately low-rank/low-dimensional. This dimension $k$ is consistent across layers, and can be interpreted as being equal to the so-called Bottleneck rank of the learned function. This structure has been shown to extend to CNNs in [22], and we will observe a similar structure in our leaky ResNets, further showcasing its generality.

More generally, our goal is to describe the 'representation geodesics' of DNNs: the paths in representation space from input to output representation. The advantage of ResNets (leaky or not) over FCNNs is that these geodesics can be approximated by continuous paths and are described by differential equations (as described by the Hamiltonian reformulation). By decomposing the Hamiltonian, we observe a separation of timescales for large depths, with slow layers with low COI/dimension, and fast layers with high COI/dimension.

### 3.1. Separation of Timescales

If $\mathrm{Im}A_p^T \subset \mathrm{Im}\sigma(A_p)^T$, we plug in $B_p = \tilde{L}^{-1}W_p\sigma(A_p)^+ = (A_p + \tilde{L}^{-1}\partial_p A_p)K_p^+$ inside equation 1 and obtain that the Hamiltonian equals the sum of the kinetic and potential energies:

$$\mathcal{H} = \frac{\tilde{L}}{2}\left\|A_p + \tilde{L}^{-1}\partial_p A_p\right\|_{K_p}^2 - \tilde{L}\left\langle A_p + \tilde{L}^{-1}\partial_p A_p, A_p\right\rangle_{K_p} = \frac{1}{2\tilde{L}}\|\partial_p A_p\|_{K_p}^2 - \frac{\tilde{L}}{2}\|A_p\|_{K_p}^2. \tag{2}$$

Since $\|\partial_p A_p\|_{K_p} = \tilde{L}\sqrt{\|A_p\|^2_{K_p} + \frac{2}{\tilde{L}}\mathcal{H}}$, for large $\tilde{L}$, the derivative $\partial_p A_p$ is only finite at $p$s where the COI $\|A_p\|^2_{K_p}$ is close to $-\frac{2}{\tilde{L}}\mathcal{H}$. On the other hand, $\partial_p A_p$ will blow up for all $p$ with a finite gap $\sqrt{\|A_p\|^2_{K_p} + \frac{2}{\tilde{L}}\mathcal{H}} > 0$ between the COI and the Hamiltonian. This suggests a separation of timescales as $\tilde{L} \to \infty$, with slow dynamics ($\|\partial_p A_p\|_{K_p} \sim 1$) in layers whose COI/dimension is close to $-\frac{2}{\tilde{L}}\mathcal{H}$ and fast dynamics ($\|\partial_p A_p\|_{K_p} \sim \tilde{L}$) in the high COI/dimension layers.

But the assumption $\mathrm{Im}A_p^T \subset \mathrm{Im}\sigma(A_p)^T$ seems to rarely be true in practice, and both kinetic and COI are often infinite in practice, canceling each other to produce a finite Hamiltonian. This means that the separation of timescales argument presented in the preceding paragraph needs to be adapted to avoid these explosions. We solve this issue by defining the $\gamma$-stable kinetic energy $\frac{1}{2\tilde{L}}\|\partial_p A_p\|^2_{(K_p+\gamma I)}$ and potential energy $\frac{\tilde{L}}{2}\|A_p\|^2_{(K_p+\gamma I)}$. These energies remain bounded as long as $\gamma$ is not too small, and their sum equals the $\gamma$-Hamiltonian $\mathcal{H}_{\gamma,p}$ which is close to the true Hamiltonian $\mathcal{H}$ as long as $\gamma$ is small enough. This allows us to translate the argument presented in the previous paragraph into a formal statement (up to approximation errors):

**Theorem 4.** *For any geodesic, we have*

$$\mathcal{H} = \frac{1}{2\tilde{L}}\left\|\partial_p A_p + \gamma\tilde{L}B_p\right\|^2_{(K_p+\gamma I)} - \frac{\tilde{L}}{2}\|A_p\|^2_{(K_p+\gamma I)} - \gamma\frac{\tilde{L}}{2}\|B_p\|^2.$$

*Therefore if $\|B_p\| \leq c$, we can bound the distance between the Hamiltonians*

$$\left|\frac{2}{\tilde{L}}\mathcal{H} - \frac{2}{\tilde{L}}\mathcal{H}_\gamma\right| \leq \frac{2}{\tilde{L}}\|\partial_p A_p\|_{(K_p+\gamma I)}\sqrt{\gamma}c + \gamma c^2 \tag{3}$$

*and guarantee that the rate of change $\partial_p A_p$ scales with $\tilde{L}$ times the extra-dimensionality*

$$\left|\|\partial_p A_p\|_{(K_p+\gamma I)} - \tilde{L}\sqrt{\|A_p\|^2_{(K_p+\gamma I)} + \frac{2}{\tilde{L}}\mathcal{H}}\right| \leq 2\tilde{L}\sqrt{\gamma}c.$$

*Finally we can guarantee that the rescaled Hamiltonian $-\frac{2}{\tilde{L}}\mathcal{H}$ approaches the minimal $\gamma$-COI from below as $\tilde{L} \to \infty$ (up to $\gamma c^2$ terms):*

$$-\left(\frac{1}{\tilde{L}}\ell_{\gamma,\tilde{L}} + \sqrt{\gamma}c\right)^2 \leq -\frac{2}{\tilde{L}}\mathcal{H} - \min_p\left\|A_p^{\tilde{L}}\right\|^2_{(K_p+\gamma I)} \leq \gamma c^2, \tag{4}$$

*for the path length $\ell_{\gamma,\tilde{L}} = \int_0^1 \left\|\partial_p A_p^{\tilde{L}}\right\|_{(K_p+\gamma I)} dp$.*

In practice, the size of $\|B_p\|^2$ can vary a lot throughout the layers, we therefore suggest choosing a $p$-dependent $\gamma$ (the proof of Theorem 4 can directly be extended to allow for this dependence): $\gamma_p = \gamma_0\|\sigma(A_p)\|^2_{op} = \gamma_0\|K_p\|^2_{op}$. There are two motivations for this: first it is natural to have $\gamma$ scale with $K_p$,; and second, since $W_p = \tilde{L}B_p\sigma(A_p)^T$ is of approximately constant size (thanks to balancedness, see Appendix A.3), we typically have that the size of $B_p$ is inversely proportional to that of $\sigma(A_p)$, so that $\gamma_p\|B_p\|^2$ should remain roughly the same size for all $p$.

Theorem 4 first shows that distance between the Hamiltonian and stable Hamiltonian is small in comparison to scale of the kinetic and potential energies (indeed the term $\frac{2}{\tilde{L}}\|\partial_p A_p\|_{(K_p+\gamma I)}\sqrt{\gamma}c$ can be large if the kinetic energy is large, but it will remain small in comparison to the kinetic energy). Since the kinetic energy can be approximated by the Hamiltonian minus the potential energy, the norm of the derivative $\|\partial_p A_p\|_{(K_p+\gamma i)}$ is close to $\tilde{L}$ times the 'extra-COI' $\sqrt{\|A_p\|^2_{(K_p+\gamma I)} + \frac{2}{\tilde{L}}\mathcal{H}} \approx \sqrt{\|A_p\|^2_{(K_p+\gamma I)} - \min_q\|A_q\|^2_{(K_q+\gamma I)}}$ (where the approximation of the Hamiltonian by the minimal COI for large $\tilde{L}$ comes from equation 4), which describes the separation of timescales, with slow ($\|\partial_p A_p\|_{K_p+\gamma I} \sim 1$) dynamics at layers $p$ where the COI is almost optimal and fast ($\|\partial_p A_p\|_{K_p+\gamma I} \sim \tilde{L}$) dynamics everywhere the COI is far from optimal.

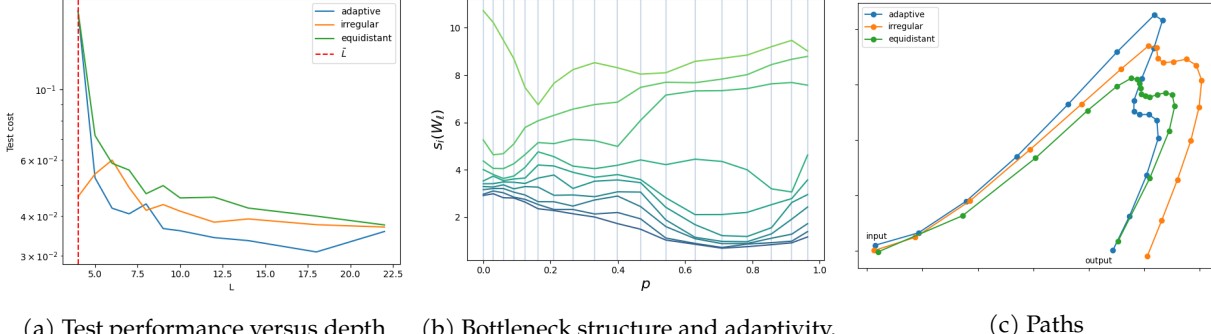

(a) Test performance versus depth    (b) Bottleneck structure and adaptivity.    (c) Paths

Figure 2: **Discretization:** We train networks with a fixed $\tilde{L} = 3$ over a range of depths $L$ and definitions of $\rho_\ell$s. The true function $f^* : \mathbb{R}^{30} \to \mathbb{R}^{30}$ is the composition of three random ResNets $g_1, g_2, g_3$ mapping from dim. 30 to 6 to 3 to 30. (a) Test error as a function of $L$ for different discretization schemes. (b) Weight spectra across layers for adaptive $\rho_\ell$ ($L = 18$), grey vertical lines represents the steps $p_\ell$. The Bottleneck structure is more complex for this task, reflecting the more complex true function $f^* = g_3 \circ g_2 \circ g_1$: we see a dimension 3 bottleneck around $p = 0.7$, but the network remains at an intermediate dimension aroung $p = 0.3$, perhaps reflecting the intermediate dimension of 6 in the true function. (c) 2D projection of the representation paths $A_p$ for $L = 18$. Observe how adaptive $\rho_\ell$s appears to better spread out the steps.

Assuming a finite length $\ell_{\gamma,\tilde{L}} < \infty$, the norm of the derivative must be finite at almost all layers, which is only possible if the COI/dimensionality is optimal in almost all layers, with only a countable number of short high COI/dimension jumps. Empirically, we observe that these jumps typically appear at the beginning and end of the network, because the input and output dimensionality where the COI is fixed and thus non-optimal in general. This explains the fast regions close to the beginning and end of the network. We have actually never observed any jump in the middle of the network, though we are not able to rule them out theoretically.

If we assume that the paths $A_p$ are stable under adding a neuron, then we can additionally guarantee that the representations in the slow layers ('inside the Bottleneck') will be non-negative:

**Proposition 5.** *Let $A_p^{\tilde{L}}$ be a uniformly bounded sequence of local minima for increasing $\tilde{L}$, at any $p_0 \in (0, 1)$ such that $\|\partial_p A_p\|$ is uniformly bounded in a neighborhood of $p_0$ for all $\tilde{L}$, then $A_{p_0}^\infty = \lim_{\tilde{L}} A_{p_0}^{\tilde{L}}$ is non-negative if it exists.*

We therefore know that the optimal COI $\min_q \|A_q\|^2_{(K_q+\gamma I)}$ is close to the dimension of the limiting representations $A_{p_0}^\infty$, i.e. it must be an integer $k^*$ which we call the Bottleneck rank of the sequence of minima since it is closely related to the Bottleneck rank introduced in [20]. The Hamiltonian $\mathcal{H}$ is then close to $-\frac{\tilde{L}}{2}k^*$.

Figure 1 illustrates these phenomena: the unstable and stable Hamiltonians approach the rank $k^* = 5$ from below, while the minimal COI approaches it from above; The kinetic energy is proportional to the extra COI, and they are both large towards the beginning and end of the network where the weights $W_p$ are higher dimensional. We see in Figure 1c that the hamiltonian and stable Hamiltonian are not exactly constant, but it still varies significantly less than the kinetic and potential energies.

Because of the non-convexity of the loss we are considering, there are likely distinct sequences of local minima as $\tilde{L} \to \infty$ of different ranks, depending on what low-dimension they reach inside their bottleneck. Indeed in our experiments we have seen that the number of dimensions that are kept inside the bottleneck can vary by 1 or 2, and in FCNN distinct sequences of depth increasing minima with different ranks have been observed in [21].

# 4. Discretization Scheme

To use such Leaky ResNets in practice, we need to discretize over the range $[0, 1]$. For this we choose a set of layer-steps $\rho_1, \ldots, \rho_L$ with $\sum \rho_\ell = 1$, and define the activations at the locations $p_\ell = \rho_1 + \cdots + \rho_\ell \in [0, 1]$ recursively as

$$\alpha_{p_0}(x) = x$$
$$\alpha_{p_\ell}(x) = (1 - \rho_\ell \tilde{L})\alpha_{p_{\ell-1}}(x) + \rho_\ell W_{p_\ell} \sigma\left(\alpha_{p_{\ell-1}}(x)\right)$$

and the regularized cost $\mathcal{L}(\theta) = C(\alpha_1(X)) + \frac{\lambda}{2\tilde{L}}\sum_{\ell=1}^{L}\rho_\ell\|W_{p_\ell}\|^2$, for the parameters $\theta = (W_{p_1}, \ldots, W_{p_L})$. Note that it is best to ensure that $\rho_\ell \tilde{L}$ remains smaller than $1$ so that the prefactor $(1 - \rho_\ell \tilde{L})$ does not become negative, though we will also discuss certain setups where it might be okay to take larger layer-steps.

Now comes the question of how to choose the $\rho_\ell$s. We consider three options:

**Equidistant:** The simplest choice is to choose equidistant points $\rho_\ell = \frac{1}{L}$. Note that the condition $\rho_\ell \tilde{L} < 1$ then becomes $L > \tilde{L}$. But this choice might be ill adapted in the presence of a Bottleneck structure due to the separation of timescales.

**Irregular:** Since we typically observe that the fast layers appear close to the inputs and outputs with a slow bottleneck in the middle, one could simply choose the $\rho_\ell$ to be go from small to large and back to small as $\ell$ ranges from $1$ to $L$. This way there are many discretized layers in the fast regions close to the input and output and not too many layers inside the Bottleneck where the representations are changing less. More concretely one can choose $\rho_\ell = \frac{1}{L} + \frac{a}{L}(\frac{1}{4} - \left|\frac{\ell}{L} - \frac{1}{2}\right|)$ for $a \in [0, 1)$, the choice $a = 0$ leads to an equidistant mesh, but increasing $a$ will lead to more points close to the inputs and outputs. To guarantee $\rho_\ell \tilde{L} < 1$, we need $L > (1 + a\frac{1}{4})\tilde{L}$.

**Adaptive:** This can be further improved by using adaptive $\rho$s. Ideally we would like to choose $\rho_\ell = \frac{c_\ell}{\sum_k c_k}$ for $c_\ell = \|A_{p_\ell}\|/\|\partial_p A_{p_\ell}\|$ to guarantee that the distances $\left\|A_{p_\ell} - A_{p_{\ell-1}}\right\|/\|A_{p_\ell}\|$ are approximately the same for all $\ell$. Approximating the derivative with a finite difference, we update $\rho_\ell \leftarrow \frac{\tilde{c}_\ell}{\sum_k \tilde{c}_k}$ for $\tilde{c}_{p_\ell} = \rho_\ell\|A_{p_\ell}\|/\|A_{p_\ell} - A_{p_{\ell-1}}\|$ every few training steps. For large networks, this has negligible computational cost (an approx. 2% longer training time in some experiments).

Figure 2 illustrates the effect of the choice of $\rho_\ell$ for different depths $L$, we see a small but consistent advantage in the test error when using adaptive or irregular $\rho_\ell$s. Looking at the resulting Bottleneck structure, we see that the adaptive $\rho_\ell$s adapts to the bottleneck structure, putting more steps in the first and last layers, and less in the low-dimensional middle layers.

# 5. Conclusion

We have described the representation geodesics $A_p$ of Leaky ResNets and decomposed the Hamiltonian as the sum of a kinetic and potential energy, where the kinetic energy measures the size of the derivative $\partial_p A_p$, while the potential energy is inversely proportional to the cost of identity, which is a measure of dimensionality of the representations. As the effective depth of the network grows, the potential energy dominates and we observe a separation of timescales that leads to a Bottleneck structure.

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

# A. Proofs

## A.1. Cost of Identity

Here are the proofs for the two Propositions of section 2.3.

**Proposition 6** (Proposition 2 in the main). *A local minimum of $A \mapsto \|A\|_K^2$ is said to be stable if it remains a local minimum after concatenating a zero vector $A' = \begin{pmatrix} A \\ 0 \end{pmatrix} \in \mathbb{R}^{(w+1)\times N}$. All stable minima are non-negative, and satisfy $\|A\|_K^2 = \|A\|_{\bar{K}}^2 = \mathrm{Rank}A$.*

*Proof.* At a critical point of the COI with bias $A \mapsto \|A\|_K^2$, the derivative w.r.t. to scaling the representation $A$ up must be zero, i.e.

$$0 = \partial_s \mathrm{Tr}\left[ s^2 A^T A \left( s^2 \bar{K} + \mathbf{1}_N \mathbf{1}_N^T \right)^+ \right]\Big|_{s=1}$$
$$= 2\mathrm{Tr}\left[ A^T A K^+ \right] - 2\mathrm{Tr}\left[ A^T A K^+ \bar{K} K^+ \right]$$
$$= 2\mathbf{1}_N^T K^+ A^T A K^+ \mathbf{1}_N,$$

which implies that $A K^+ \mathbf{1}_N = 0$.

Furthermore, since $A$ is a stable minima, the COI of the nearby point $\begin{pmatrix} A \\ \epsilon z \end{pmatrix}$ for $z \in \mathrm{Im}\sigma(A)^T$

$$\mathrm{Tr}\left[ \left( A^T A + \epsilon^2 z z^T \right) \left( K + \epsilon^2 \bar{\sigma}(z) \bar{\sigma}(z)^T \right)^+ \right] = \|A\sigma(A)^+\|^2 + \epsilon^2 \|z^T \sigma(A)^+\|^2 - \epsilon^2 \|\bar{\sigma}(z)^T K^+ A^T\|^2 + O(\epsilon^4),$$

must not be smaller than $\|A\sigma(A)^+\|^2$ for small $\epsilon$. This implies that

$$z^T K^+ z = \|z^T \sigma(A)^+\|^2 \geq \|\bar{\sigma}(z)^T K^+ A^T\|^2 = \bar{\sigma}(z)^T K^+ A^T A K^+ \bar{\sigma}(z).$$

Let us now choose $z = \bar{K}_i = K_i - \mathbf{1}_N$, which has positive entries so that $\bar{\sigma}(\bar{K}_i) = \bar{K}_i$ and

$$\bar{K}_i^T K^+ \bar{K}_i^T \geq \bar{K}_i^T K^+ A^T A K^+ \bar{K}_i.$$

Both sides can be simplified:

$$\bar{K}_i^T K^+ \bar{K}_i^T = \|\sigma(A_i)\|^2 - 2K_i^T K^+ \mathbf{1}_N + \mathbf{1}_N K^+ \mathbf{1}_N = \|\sigma(A_i)\|^2 - 2 + \mathbf{1}_N K^+ \mathbf{1}_N$$

since $K_i^T K^+ \mathbf{1}_N = e_i P_{\mathrm{Im}K} \mathbf{1}_N = e_i \mathbf{1}_N = 1$ because $\mathbf{1}_N$ lies in the image of $K$; and since $A K^+ \mathbf{1}_N = 0$

$$\bar{K}_i^T K^+ A^T A K^+ \bar{K}_i = \|A_i\|^2 - 2K_i^T K^+ A^T A K^+ \mathbf{1}_N + \mathbf{1}_N^T K^+ A^T A K^+ \mathbf{1}_N = \|A_i\|^2.$$

This implies that

$$\|\sigma(A_i)\|^2 - 2 + \mathbf{1}_N K^+ \mathbf{1}_N \geq \|A_i\|^2.$$

But we have $\mathbf{1}_N K^+ \mathbf{1}_N \leq 1$ since

$$\mathbf{1}_N K^+ \mathbf{1}_N = \lim_{\gamma \searrow 0} \mathbf{1}_N \left( K + \gamma I \right)^{-1} K \left( K + \gamma I \right)^{-1} \mathbf{1}_N$$
$$\leq \lim_{\gamma \searrow 0} \mathbf{1}_N \left( K + \gamma I \right)^{-1} \left( K + \gamma I \right) \left( K + \gamma I \right)^{-1} \mathbf{1}_N$$
$$= \lim_{\gamma \searrow 0} \mathbf{1}_N \left( K + \gamma I \right)^{-1} \mathbf{1}_N,$$

and by Shermann-Morrison formula:

$$\mathbf{1}_N \left( \bar{K} + \mathbf{1}_N \mathbf{1}_N^T + \gamma I \right)^{-1} \mathbf{1}_N = \mathbf{1}_N \left( \bar{K} + \gamma I \right)^{-1} \mathbf{1}_N - \frac{\left( \mathbf{1}_N \left( \bar{K} + \gamma I \right)^{-1} \mathbf{1}_N \right)^2}{1 + \mathbf{1}_N \left( \bar{K} + \gamma I \right)^{-1} \mathbf{1}_N} = \frac{\mathbf{1}_N \left( \bar{K} + \gamma I \right)^{-1} \mathbf{1}_N}{1 + \mathbf{1}_N \left( \bar{K} + \gamma I \right)^{-1} \mathbf{1}_N} \leq 1,$$

with equality if and only if $\lim_{\gamma \searrow 0} \mathbf{1}_N \left( \bar{K} + \gamma I \right)^{-1} \mathbf{1}_N = \infty$ which happens when $\mathbf{1}_N$ does not lie in the image of $\bar{K}$.

This leads to the bound $\|\sigma(A_i)\|^2 - 1 \geq \|A_i\|^2 + 1$, but in the other direction, we know $\|\sigma(A_i)\|^2 \leq \|A_i\|^2 + 1$, with equality if and only if $A_i$ has non-negative entries, since the ReLU satisfies $|\sigma(x)| \leq |x|$ with equality on non-negative $x$. This implies that $A_i$ has non-negative entries, and that $\mathbf{1}_N K^+ \mathbf{1}_N = 1$.

Furthermore, we have $\|A\|_K^2 \leq \|A\|_{\bar{K}}^2$ and

$$
\begin{aligned}
\|A\|_K^2 &= \lim_{\gamma \searrow 0} \operatorname{Tr}\left[\bar{K}(\bar{K} + \mathbf{1}_N \mathbf{1}_N + \gamma I)^{-1}\right] \\
&= \lim_{\gamma \searrow 0} \|A\|_{\bar{K}+\gamma I}^2 - \frac{\mathbf{1}_N(\bar{K}+\gamma I)^{-1}\bar{K}(\bar{K}+\gamma I)^{-1}\mathbf{1}_N}{1 + \mathbf{1}_N^T(\bar{K}+\gamma I)^{-1}\mathbf{1}_N^T} \\
&\geq \lim_{\gamma \searrow 0} \|A\|_{\bar{K}+\gamma I}^2 - \frac{\mathbf{1}_N(\bar{K}+\gamma I)^{-1}\mathbf{1}_N}{1 + \mathbf{1}_N^T(\bar{K}+\gamma I)^{-1}\mathbf{1}_N^T} \\
&= \|A\|_{\bar{K}}^2,
\end{aligned}
$$

since $\lim_{\gamma \searrow 0} \mathbf{1}_N \left(\bar{K}+\gamma I\right)^{-1}\mathbf{1}_N = \infty$ because $\mathbf{1}_N K^+ \mathbf{1}_N = 1$. Therefore

$$
\|A\|_K^2 = \|A\|_{\bar{K}}^2 = \operatorname{Rank} A.
$$

$\square$

**Proposition 7** (Proposition 3 in the main.). *If $w > N(N+1)$ then if $\hat{A} \in \mathbb{R}^{w \times N}$ is local minimum of $A \mapsto \|A\sigma(A)^+\|_F^2$ that is not non-negative, then there is a continuous path $A_t$ of constant COI such that $A_0 = \hat{A}$ and $A_1$ is a saddle.*

*Proof.* The local minimum $\hat{A}$ leads to a pair of $N \times N$ covariance matrices $\hat{K} = \hat{A}^T \hat{A}$ and $\hat{K}^\sigma = \sigma(\hat{A})^T \sigma(\hat{A})$. The pair $(\hat{K}, \hat{K}^\sigma)$ belongs to the conical hull $\operatorname{Cone}\left\{(\hat{A}_{i.}\hat{A}_{i.}^T, \sigma(\hat{A}_{i.})\sigma(\hat{A}_{i.})^T) : i = 1, \ldots, w\right\}$. Since this cone lies in a $N(N+1)$-dimensional space (the space of pairs of symmetric $N \times N$ matrices), we know by Caratheodory's theorem (for convex cones) that there is a conical combination $(\hat{K}, \hat{K}^\sigma - \beta^2 \mathbf{1}_{N \times N}) = \sum_{i=1}^w a_i(\hat{A}_{i.}\hat{A}_{i.}^T, \sigma(\hat{A}_{i.})\sigma(\hat{A}_{i.})^T)$ such that no more than $N(N+1)$ of the coefficients are non-zero. We now define $A_t$ to have lines $A_{t,i.} = \sqrt{(1-t) + t a_i}\hat{A}_{i.}$, so that $A_{t=0} = \hat{A}$ and at $t = 1$ at least one line of $A_{t=1}$ is zero (since at least one of the $a_i$s is zero). First note that the covariance pairs remain constant over the path: $K_t = A_t^T A_t = \sum_{i=1}^w ((1-t) + t a_i)\hat{A}_{i.}\hat{A}_{i.}^T = (1-t)\hat{K} + t\hat{K} = \hat{K}$ and similarly $K_t^\sigma = \hat{K}^\sigma$, which implies that the cost $\|A_t \sigma(A_t)^+\|_F^2 = \operatorname{Tr}\left[K_t K_t^{\sigma+}\right]$ is constant too. Second, since a representation $A$ is non-negative iff the covariances satisfy $K = K^\sigma$, the representation path $A_t$ cannot be non-negative either since it has the same kernel pairs $(\hat{K}, \hat{K}^\sigma)$ with $\hat{K} \neq \hat{K}^\sigma$.

Now (the converse of) Proposition 2 tells us that if $A_{t=1}$ is not non-negative and has a zero line, then it is not a local minimum, which implies that it is a saddle. $\square$

## A.2. Bottleneck

**Theorem 8** ( Theorem 4 in the main). *For any geodesic, we have*

$$
\mathcal{H} = \frac{1}{2\tilde{L}}\left\|\partial_p A_p + \gamma \tilde{L} B_p\right\|_{(K_p + \gamma I)}^2 - \frac{\tilde{L}}{2}\|A_p\|_{(K_p + \gamma I)}^2 - \gamma \frac{\tilde{L}}{2}\|B_p\|^2.
$$

*Therefore if $\|B_p\| \leq c$, we can bound the distance between the Hamiltonians*

$$
\left|\frac{2}{\tilde{L}}\mathcal{H} - \frac{2}{\tilde{L}}\mathcal{H}_\gamma\right| \leq \frac{2}{\tilde{L}}\|\partial_p A_p\|_{(K_p + \gamma I)}\sqrt{\gamma}c + \gamma c^2
$$

*and guarantee that the rate of change $\partial_p A_p$ scales with $\tilde{L}$ times the extra-dimensionality*

$$
\left|\|\partial_p A_p\|_{(K_p + \gamma I)} - \tilde{L}\sqrt{\|A_p\|_{(K_p + \gamma I)}^2 + \frac{2}{\tilde{L}}\mathcal{H}}\right| \leq 2\tilde{L}\sqrt{\gamma}c.
$$

*Finally we can guarantee that the rescaled Hamiltonian $-\frac{2}{\tilde{L}}\mathcal{H}$ approaches the minimal $\gamma$-COI from below as $\tilde{L} \to \infty$ (up to $\gamma c^2$ terms):*

$$-\left(\frac{1}{\tilde{L}}\ell_{\gamma,\tilde{L}} + \sqrt{\gamma}c\right)^2 \leq -\frac{2}{\tilde{L}}\mathcal{H} - \min_p \left\|A_p^{\tilde{L}}\right\|_{(K_p+\gamma I)}^2 \leq \gamma c^2,$$

*for the path length $\ell_{\gamma,\tilde{L}} = \int_0^1 \left\|\partial_p A_p^{\tilde{L}}\right\|_{(K_p+\gamma I)} dp$.*

*Proof.* (1) Since $B_p = (\frac{1}{\tilde{L}}\partial_p A_p + A_p)K_p^+$, we have

$$\left\|\frac{1}{\tilde{L}}\partial_p A_p + \gamma B_p\right\|_{(K_p+\gamma I)}^2 = \|B_p(K_p + \gamma) - A_p\|_{(K_p+\gamma I)}^2$$

$$= \left\|B_p\sigma(A_p)^T\right\|^2 + \gamma\|B_p\|^2 - 2\text{Tr}\left[B_p A_p^T\right] + \|A_p\|_{(K_p+\gamma I)}^2$$

$$= \frac{2}{\tilde{L}}\mathcal{H} + \gamma\|B_p\|^2 + \|A_p\|_{(K_p+\gamma I)}^2$$

and thus we have

$$-\frac{2}{\tilde{L}}\mathcal{H} = \|A_p\|_{(K_p+\gamma I)}^2 - \left\|\frac{1}{\tilde{L}}\partial_p A_p + \gamma B_p\right\|_{(K_p+\gamma I)}^2 + \gamma\|B_p\|^2.$$

(2) We can further simplify the previous equality to

$$\frac{2}{\tilde{L}}\mathcal{H} = \left\|\frac{1}{\tilde{L}}\partial_p A_p + \gamma B_p\right\|_{(K_p+\gamma I)}^2 - \|A_p\|_{(K_p+\gamma I)}^2 - \gamma\|B_p\|^2$$

$$= \frac{2}{\tilde{L}}\mathcal{H}_{\gamma,p} + \frac{2\gamma}{\tilde{L}}\langle\partial_p A_p, B_p\rangle_{(K_p+\gamma I)} + \gamma^2\|B_p\|_{(K_p+\gamma I)}^2 - \gamma\|B_p\|^2$$

Leading to the upper bound

$$\frac{2}{\tilde{L}}\mathcal{H} - \frac{2}{\tilde{L}}\mathcal{H}_{\gamma,p} \leq \frac{2\gamma}{\tilde{L}}\|\partial_p A_p\|_{(K_p+\gamma I)}\|B_p\|_{(K_p+\gamma I)} + \gamma^2\|B_p\|_{(K_p+\gamma I)}^2$$

$$\leq \frac{2}{\tilde{L}}\|\partial_p A_p\|_{(K_p+\gamma I)}\sqrt{\gamma}\|B_p\| + \gamma\|B_p\|^2$$

$$\leq \frac{2}{\tilde{L}}\|\partial_p A_p\|_{(K_p+\gamma I)}\sqrt{\gamma}c + \gamma c^2$$

and lower bound

$$\frac{2}{\tilde{L}}\mathcal{H} - \frac{2}{\tilde{L}}\mathcal{H}_{\gamma,p} \geq -\frac{2\gamma}{\tilde{L}}\|\partial_p A_p\|_{(K_p+\gamma I)}\|B_p\|_{(K_p+\gamma I)} - \gamma\|B_p\|^2$$

$$\geq -\frac{2}{\tilde{L}}\|\partial_p A_p\|_{(K_p+\gamma I)}\sqrt{\gamma}c - \gamma c^2.$$

(3) We have the lower bound

$$\frac{1}{\tilde{L}}\|\partial_p A_p\|_{(K_p+\gamma I)} \geq \left\|\frac{1}{\tilde{L}}\partial_p A_p + \gamma B_p\right\|_{(K_p+\gamma I)} - \|\gamma B_p\|_{(K_p+\gamma I)}$$

$$\geq \sqrt{\|A_p\|_{(K_p+\gamma I)}^2 + \frac{2}{\tilde{L}}\mathcal{H} + \gamma\|B_p\|^2} - \sqrt{\gamma}c$$

$$\geq \sqrt{\|A_p\|_{(K_p+\gamma I)}^2 + \frac{2}{\tilde{L}}\mathcal{H}} - \sqrt{\gamma}c, \tag{5}$$

and upper bound

$$\frac{1}{\tilde{L}}\left\|\partial_p A_p\right\|_{(K_p+\gamma I)} \leq \left\|\frac{1}{\tilde{L}}\partial_p A_p + \gamma B_p\right\|_{(K_p+\gamma I)} + \left\|\gamma B_p\right\|_{(K_p+\gamma I)}$$

$$\leq \sqrt{\left\|A_p\right\|_{(K_p+\gamma I)}^2 + \frac{2}{\tilde{L}}\mathcal{H}} + \gamma\left\|B_p\right\|^2 + \sqrt{\gamma}c$$

$$\leq \sqrt{\left\|A_p\right\|_{(K_p+\gamma I)}^2 + \frac{2}{\tilde{L}}\mathcal{H}} + \sqrt{\gamma}\left\|B_p\right\| + \sqrt{\gamma}c$$

$$\leq \sqrt{\left\|A_p\right\|_{(K_p+\gamma I)}^2 + \frac{2}{\tilde{L}}\mathcal{H}} + 2\sqrt{\gamma}c.$$

(4) The upper bound $-\frac{2}{\tilde{L}}\mathcal{H} - \min_p\left\|A_p^{\tilde{L}}\right\|_{(K_p+\gamma I)}^2 \leq \gamma c^2$ follows from the fact that $\left\|B_p\right\|^2 \leq c^2$. For the lower bound, we integrate the lower bound from equation 5:

$$\frac{1}{\tilde{L}}\ell_{\gamma,\tilde{L}} = \frac{1}{\tilde{L}}\int_0^1 \left\|\partial_p A_p\right\|_{(K_p+\gamma I)}dp$$

$$\geq \int_0^1 \sqrt{\left\|A_p\right\|_{(K_p+\gamma I)}^2 + \frac{2}{\tilde{L}}\mathcal{H}}dp - \sqrt{\gamma}c$$

$$\geq \sqrt{\min_p\left\|A_p\right\|_{(K_p+\gamma I)}^2 + \frac{2}{\tilde{L}}\mathcal{H}} - \sqrt{\gamma}c.$$

$\square$

**Proposition 9** (Proposition 5 in the main.). *Let $A_p^{\tilde{L}}$ be a uniformly bounded sequence of local minima for increasing $\tilde{L}$, at any $p_0 \in (0,1)$ such that $\left\|\partial_p A_p\right\|$ is uniformly bounded in a neighborhood of $p_0$ for all $\tilde{L}$, then $A_{p_0}^\infty = \lim_{\tilde{L}} A_{p_0}^{\tilde{L}}$ is non-negative.*

*Proof.* Given a path $A_p$ with corresponding weight matrices $W_p$ corresponding to a width $w$, then $\begin{pmatrix} A \\ 0 \end{pmatrix}$ is a path with weight matrix $\begin{pmatrix} W_p & 0 \\ 0 & 0 \end{pmatrix}$. Our goal is to show that for sufficiently large depths, one can under certain assumptions slightly change the weights to obtain a new path with the same endpoints but a slightly lower loss, thus ensuring that if certain assumptions are not satisfied then the path cannot be locally optimal.

Let us assume that $\left\|\partial_p A_p\right\| \leq c_1$ in a neighborhood of a $p_0 \in (0,1)$, and assume by contradiction that there is an input index $i = 1, \ldots, N$ such that $A_{p_0,\cdot i}$ has at least one negative entry, and therefore $\left\|A_{p_0,\cdot i}\right\|^2 - \left\|\sigma(A_{p_0,\cdot i})\right\|^2 = c_0 > 0$ for all $\tilde{L}$.

We now consider the new weights

$$\begin{pmatrix} W_p - \tilde{L}\epsilon^2 t(p)A_{p,\cdot i}\sigma(A_{p,\cdot i})^T & \epsilon\tilde{L}t(p)A_{p,\cdot i} \\ \epsilon\tilde{L}t(p)\sigma(A_{p,\cdot i}) & 0 \end{pmatrix}$$

for $t(p) = \max\{0, 1 - \frac{|p-p_0|}{r}\}$ a triangular function centered in $p_0$ and for an $\epsilon > 0$.

For $\epsilon$ and $r$ small enough, the parameter norm will decrease:

$$\int_0^1 \left\|\begin{matrix} W_p - \tilde{L}\epsilon^2 t(p)A_{p,\cdot i}\sigma(A_{p,\cdot i})^T & \epsilon\tilde{L}t(p)A_{p,\cdot i} \\ \epsilon\tilde{L}t(p)\sigma(A_{p,\cdot i}) & 0 \end{matrix}\right\|^2 dp$$

$$= \int_0^1 \left\|W_p\right\|^2 + \tilde{L}^2\epsilon^2 t(p)^2 \left(-\frac{2}{\tilde{L}}A_{p,\cdot i}^T W_p\sigma(A_{p,\cdot i}) + \left\|A_{p,\cdot i}\right\|^2 + \left\|\sigma(A_{p,\cdot i})\right\|^2\right)dp + O(\epsilon^4).$$

Now since $W_p\sigma(A_{p,\cdot i}) = \partial_p A_{p,\cdot i} + \tilde{L}A_{p,\cdot i}$, this simplifies to

$$\int_0^1 \left\|W_p\right\|^2 + \tilde{L}^2\epsilon^2 t(p)^2 \left(-\left\|A_{p,\cdot i}\right\|^2 + \left\|\sigma(A_{p,\cdot i})\right\|^2 - \frac{1}{\tilde{L}}A_{p,\cdot i}^T\partial_p A_{p,\cdot i}\right)dp + O(\epsilon^4).$$

By taking $r$ small enough, we can guarantee that $-\|A_{p,\cdot i}\|^2 + \|\sigma(A_{p,\cdot i})\|^2 < -\frac{c_0}{2}$ for all $p$ such that $t(p) > 0$, and for $\tilde{L}$ large enough we can guarantee that $\left|\frac{1}{\tilde{L}}A_{p,\cdot i}^T \partial_p A_{p,\cdot i}\right|$ is smaller then $\frac{c_0}{4}$, so that we can guarantee that the parameter norm will be strictly smaller for $\epsilon$ small enough.

We will now show that with these new weights the path becomes approximately $\begin{pmatrix} A_p \\ \epsilon a_p \end{pmatrix}$ where

$$a_p = \tilde{L}\int_0^p t(q)K_{p,i\cdot}e^{\tilde{L}(q-p)}dq.$$

Note that $a_p$ is positive for all $p$ since $K_p$ has only positive entries. Also note that as $\tilde{L} \to \infty$, $a_p \to t(p)K_{p,i\cdot}$ and so that $a_0 \to 0$ and $a_1 \to 0$.

On one hand, we have the time derivative

$$\partial_p \begin{pmatrix} A_p \\ \epsilon a_p \end{pmatrix} = \begin{pmatrix} W_p\sigma(A_p) - \tilde{L}A_p \\ \epsilon\tilde{L}\left(t(p)K_{p,i\cdot} - a_p\right) \end{pmatrix}.$$

On the other hand the actual derivative as determined by the new weights:

$$\begin{pmatrix} W_p - \tilde{L}\epsilon^2 t(p)A_{p,\cdot i}\sigma(A_{p,\cdot i})^T & \epsilon\tilde{L}t(p)A_{p,\cdot i} \\ \epsilon\tilde{L}t(p)\sigma(A_{p,\cdot i}) & 0 \end{pmatrix}\begin{pmatrix} \sigma(A_p) \\ \epsilon\sigma(a_p) \end{pmatrix} - \tilde{L}\begin{pmatrix} A_p \\ \epsilon a_p \end{pmatrix}$$

$$= \begin{pmatrix} W_p\sigma(A_p) - \tilde{L}A_p - \tilde{L}\epsilon^2 t(p)^2 A_{p,\cdot i}K_{p,i\cdot} + \tilde{L}\epsilon^2 t(p)A_{p,\cdot i}a_p \\ \epsilon\tilde{L}t(p)K_{p,i\cdot} - \epsilon\tilde{L}a(p) \end{pmatrix}.$$

The only difference is the two terms

$$-\tilde{L}\epsilon^2 t(p)^2 A_{p,\cdot i}K_{i\cdot} + \tilde{L}\epsilon^2 t(p)A_{p,\cdot i}a_p = -\tilde{L}\epsilon^2 t(p)A_{p,\cdot i}\left(t(p)K_{i\cdot} - a_p\right).$$

One can guarantee with a Grönwall type of argument that the representation path resulting from the new weights must be very close to the path $\begin{pmatrix} A_p \\ \epsilon a_p \end{pmatrix}$. $\qquad\square$

## A.3. Balancedness

This paper will heavily focus on the Hamiltonian $\mathcal{H}_p$ that is constant throughout the layers $p \in [0, 1]$, and how it can be interpreted. Note that the Hamiltonian we introduce is distinct from an already known invariant, which arises as the result of so-called balancedness, which we introduce now.

Though this balancedness also appears in ResNets, it is easiest to understand in fullyconnected networks. First observe that for any neuron $i \in 1, \ldots, w$ at a layer $\ell$ one can multiply the incoming weights $(W_{\ell,i\cdot}, b_{\ell,i})$ by a scalar $\alpha$ and divide the outcoming weights $W_{\ell+1,\cdot i}$ by the same scalar $\alpha$ without changing the subsequent layers. One can easily see that the scaling that minimize the contribution to the parameter norm is such that the norm of incoming weights equals the norm of the outcoming weights $\|W_{\ell,i\cdot}\|^2 + \|b_{\ell,i}\|^2 = \|W_{\ell+1,\cdot i}\|^2$. Summing over the $i$s we obtain $\|W_\ell\|_F^2 + \|b_\ell\|^2 = \|W_{\ell+1}\|_F^2$ and thus $\|W_\ell\|_F^2 = \|W_1\|_F^2 + \sum_{k=1}^{\ell-1}\|b_k\|_F^2$, which means that the norm of the weights is increasing throughout the layers, and in the absence of bias, it is even constant.

Leaky ResNet exhibit the same symmetry:

**Proposition 10.** *At any critical $W_p$, we have $\|W_p\|^2 = \|W_0\|^2 + \tilde{L}\int_0^p \|W_{p,\cdot w+1}\|^2 dq$.*

*Proof.* This proofs handles the bias $W_{p,\cdot(w+1)}$ differently to the rest of the weights $W_{p,\cdot(1:w)}$, to simplify notations, we write $V_p = W_{p,\cdot(1:w)}$ and $b_p = W_{p,\cdot(w+1)}$ for the bias.

First let us show that choosing the weight matrices $\tilde{V}_q = r'(q)V_{r(q)}$ and bias $\tilde{b}_q = r'(q)e^{\tilde{L}(r(q)-q)}b_{r(q)}$ leads to the path $\tilde{A}_q = e^{\tilde{L}(r(q)-q)}A_{r(q)}$. Indeed the path $\tilde{A}_q = e^{\tilde{L}(r(q)-q)}A_{r(q)}$ has the right value

when $p = 0$ and it then satisfies the right differential equation:

$$\partial_q \tilde{A}_q = \tilde{L}(r'(q) - 1)\tilde{A}_q + e^{\tilde{L}(r(q)-q)}r'(q)\partial_p A_{r(q)}$$
$$= \tilde{L}(r'(q) - 1)\tilde{A}_q + e^{\tilde{L}(r(q)-q)}r'(q)\left(-\tilde{L}A_{r(q)} + V_{r(q)}\sigma(A_{r(q)}) + b_{r(q)}\right)$$
$$= -\tilde{L}\tilde{A}_q + r'(q)A_{r(q)}\sigma\left(\tilde{Z}_q\right) + e^{\tilde{L}(r(q)-q)}r'(q)b_{r(q)}$$
$$= \tilde{V}_q\sigma\left(\tilde{A}_q\right) + \tilde{b}_q - \tilde{L}\tilde{A}_q$$

The optimal reparametrization $r(q)$ is therefore the one that minimizes

$$\int_0^1 \left\|\tilde{W}_q\right\|^2 + \left\|\tilde{b}_q\right\|^2 dq = \int_0^1 r'(q)^2\left(\left\|W_{r(q)}\right\|^2 + e^{2\tilde{L}(r(q)-q)}\left\|b_{r(q)}\right\|^2\right) dq$$

For the identity reparametrization $r(q) = q$ to be optimal, we need

$$\int_0^1 2dr'(p)\left(\|W_p\|^2 + \|b_p\|^2\right) + 2\tilde{L}dr(p)\|b_p\|^2 dp = 0$$

for all $dr(q)$ with $dr(0) = dr(1) = 0$. Since

$$\int_0^1 dr'(p)\left(\|W_p\|^2 + \|b_p\|^2\right) dp = -\int_0^1 dr(p)\partial_p\left(\|W_p\|^2 + \|b_p\|^2\right) dq,$$

we need

$$\int_0^1 dr(p)\left[-\partial_p\left(\|W_p\|^2 + \|b_p\|^2\right) + \tilde{L}\|b_p\|^2\right] dp = 0$$

and thus for all $p$

$$\partial_p\left(\|W_p\|^2 + \|b_p\|^2\right) = \tilde{L}\|b_p\|^2.$$

Integrating, we obtain as needed

$$\|W_p\|^2 + \|b_p\|^2 = \|W_0\|^2 + \|b_0\|^2 + \tilde{L}\int_0^p \|b_q\|^2 dq.$$

$\square$

## B. Experimental Setup

Our experiments make use of synthetic data to train leaky ResNets so that the Bottleneck rank $k^*$ is known for our experiments. The synthetic data is generated by teacher networks for a given true rank $k^*$. To construct a bottleneck, the teacher network is a composition of networks for which the the inner-dimension is $k^*$. For data, we sampled a thousand data points for training, and another thousand for testing which are collectively augmented by demeaning and normalization.

To train the leaky ResNets, it is important for them to be wide, usually wider than the input or output dimension, we opted for a width of 200. However, the width of the representation must be constant to implement leaky residual connections, so we introduce a single linear mapping at the start, and another at the end, of the forward pass to project the representations into a higher dimension for the paths. These linear mappings can be either learned or fixed.

To achieve a tight convergence in training, we train primarily using Adam using Mean Squared Error as a loss function, and our custom weight decay function. After training on Adam (we found 20000 epochs to work well), we then train briefly (usually 10000 epochs) using SGD with a smaller learning rate to tighten the convergence.

The bottleneck structure of a trained network, as seen in Figure 3, can be observed in the spectra of the weight matrices $W_p$ at each layer. As long as the training is not over-regularized ($\lambda$ too large)

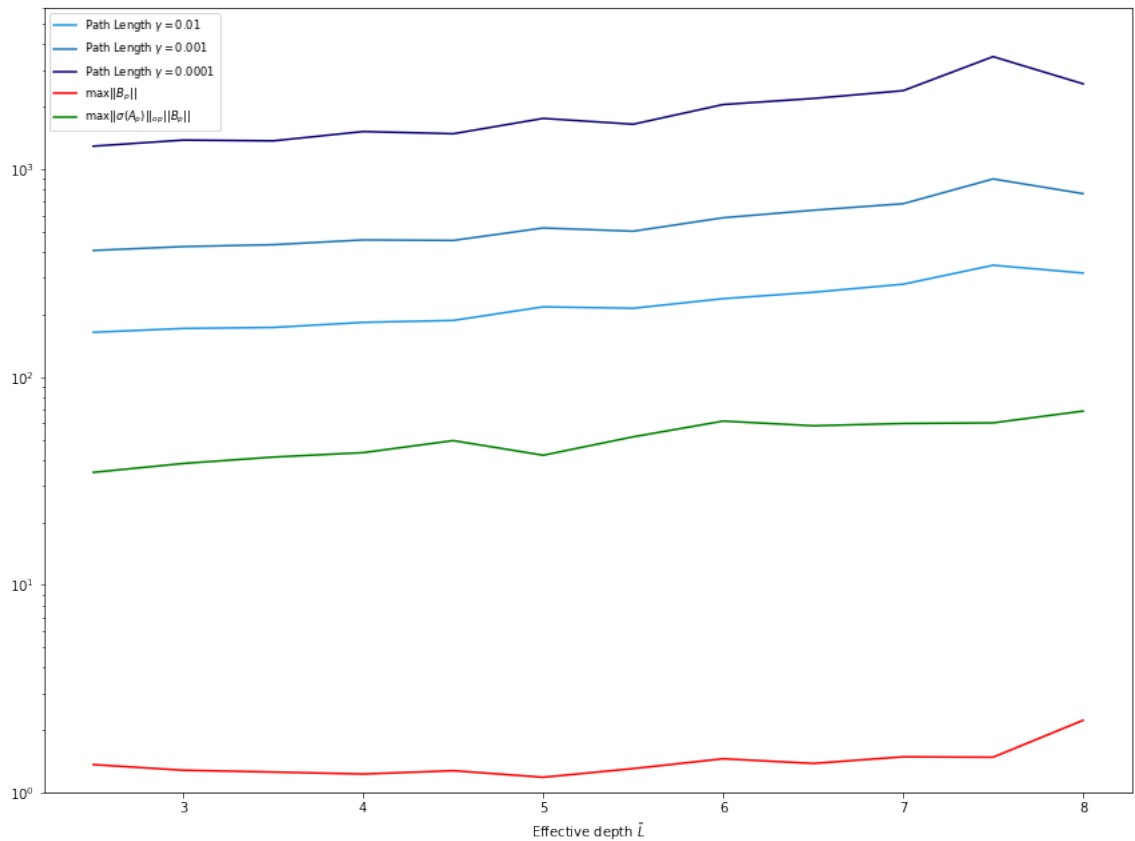

Figure 3: Various properties of the Hamiltonian dynamics of Leaky ResNets which remain bounded

then the spectra reveals a clear separation between $k^*$ number of large values as the rest decay. In our experiments, $\lambda = 0.002$ yielded good results. To facilitate the formation of the bottleneck structure, $L$ should be large, for our experiments we used $L = 50$ and then a range from 4 to 22. Figure 2a shows how larger $L$, which have better separation between large and small singular values, lead to improved test performance.

As first noted in section 2.2, solving for the Cost Of Identity, the kinetic energy, and the Hamiltonian $\mathcal{H}$ is difficult due to the instability of the pseudo-inverse. Although the relaxation $(K_p + \gamma I)$ improves the stability, we also utilize the solve function to avoid computing a pseudo-inverse altogether. The stability of these computations rely on the boundedness of some additional properties: the path length $\int ||\partial_p A_p|| \; dp$, as well as the magnitudes of $B_p$, and $B_p \sigma(A_p)^T$ from the Hamiltonian reformulation. Figure 3 shows how their respective magnitudes remains relatively constant as the effective depth $\tilde{L}$ grows.

For compute resources, these small networks are not particularly resource intensive. Even on a CPU, it only takes a couple minutes to fully train a leaky ResNet.

