# OpenReview forum: "Hamiltonian Mechanics of Feature Learning: Bottleneck Structure in Leaky ResNets"
_CPAL.cc/2025/Proceedings_Track — CPAL 2025 (Proceedings Track) Oral_

### Official Review · Reviewer_wRA6 · 2025-01-07
**Review of submission 65**

**Rating:** 7
**Confidence:** 2

**Review:**

**Strengths**

1. The paper was well motivated.

2. The need for an understanding of feature learning is important and the methods developed by the authors provide new tools for studying this question. This makes the work well aligned with the goals of CPAL.

3. The simple numerical results were effective at demonstrating the theoretical results.

4. The reformulation of the Lagranian dynamics to Hamiltonian dynamics was great and the benefit of avoiding a pseudo-inverse was clear.

**Weaknesses**

1. The main weakness to me (which could be due to the fact that I am not so familiar with the relevant literature this approach builds on) is that the big result of the paper, the "separation of timescales" was a little lost in the technical discussion. Connecting these results back to the original Bottleneck work could be helpful to the reader as well to understand the bigger picture.

2. Probably something silly I'm doing, but I don't quite see how Proposition 1 is proved. In particular, it seems like the numerator of the lower bound should be ||A||_*  and not ||A||^2_*.

3. It would be helpful to explicitly define what is meant by a "path". The authors refer to paths as both $W_p$ and $A_p$. I understand that those are related, but being more concrete early in the paper (like you are later in the paper, saying that "$p \mapsto A_p$" (line 126) ) might make it easier to follow along.

**Minor points**

1. Line 270 says $k^* = 3$ but in Fig. 1 the Hamiltonian converges to $k^* = 5$.

2. Why did the authors chose the true function $f^*$ being the composition of three random ResNets $g_1, g_2, g_3$ in Fig. 2?

3. Small typo on line 301 - should be $\rho (||A_l - A_{l - 1}|| / ||A_p||) = p_l c_l$.

4. FCNN should be defined before used.

5. I think one thing that might help the flow of the paper is if Sec. 1.1-1.4 were put into their own section separate from the Introduction.

6. Very minor, but "time scale" and "timescale" are both used

---

### Official Review · Reviewer_1ppG · 2025-01-10
**Great theoretical work showing the bottleneck structures in DNN**

**Rating:** 8
**Confidence:** 4

**Review:**

This paper novelly introduces the Leaky ResNet and analyzes the feature learning dynamics of this DNN model using a Hamiltonian reformulation. By examining the balance between kinetic and potential energy, it demonstrates that bottleneck structures emerge when potential energy becomes dominant for large depth. While the paper includes several intuitive claims and assumptions that might limit the generalizability of its propositions and theorems, I believe it offers an effective approach to understanding deep neural networks and inspires further research.

**Strengths**

1. The problem addressed is challenging, timely, and interesting. Bottleneck structures have been widely discussed and critiqued in recent years, and the Leaky ResNet provides a novel and relatively concrete way to illustrate the bottleneck structure.

2. Hamiltonian dynamics and the concept of "representation geodesics" in ResNet provide a natural and well-founded framework, offering a novel perspective on understanding feature learning in DNN.

3. The paper includes good experiments to support its theoretical claims.

4. As mentioned in Section 3, there is potential to improve network in practical, based on the insights presented in this work.

**Weaknesses and Questions**
1. The writing is very casual. I feel it is written in a style similar to lecture notes.    There are many assumptions stated at each stage of this paper. Some of these assumptions are only for intuitive understanding. When it comes to theorem and propositions, it is not very clear which assumptions are used exactly.

2. In the right-hand side of the equation between lines 114 and 115, I understand that the first and third terms play a central role, but I don’t get why the second term doesn’t. This term arises from the introduction of non-linearity, but the whole paper is based on the Leaky ReLU setting. Why isn’t it important in this model? also for finite $\tilde{L}$? Could the author elaborate further on this point?

3. Based on the insight from the separation of timescale, the authors show that a practical network with discretized layers can be improved by ensuring that the distances $\||(A_l - A_{l-1}) \||/ \||A_p\||$ are consistent throughout layers. However, why is a more uniform speed across layers directly tied to better generalization (showed in Figure 2 a)?   Is there any theoretical insight here? any assumption on the dataset? There are also some recent works showing that a uniform feature learning/neural collapse across different layers helps generalization[1,2], which might be related.

[1] He, H., & Su, W.J., 2023. A law of data separation in deep learning. Proceedings of the National Academy of Sciences, 120(36), p.e2221704120.

[2] Shi, C., Pan, L., & Dokmanić, I., 2024. A spring-block theory of feature learning in deep neural networks. arXiv preprint arXiv:2407.19353.

**Minimal Comments**
There is a typo at line 176: an extra full stop.

In general, this is a great theoretical work that offers new understanding and motivates further research.

---

### Official Review · Reviewer_NQXU · 2025-01-11

**Rating:** 8
**Confidence:** 3

**Review:**

This paper studies the representation geodesics of “Leaky ResNets” and provides qualitative and quantitative insights into their learned representations, including the emergence of a bottleneck structure (also observed in other neural networks). Overall the presentation is reasonably clear and helpful, although understandably very technical. The Lagrangian and Hamiltonian formulations are supplemented with helpful qualitative explanations and limitations in addition to the technical details.

Questions for the authors:

Could you please further clarify the role of the term \gamma, particularly as it appears in Fig. 1 and in the theory? (This appears to be the point of Theorem 4, but may be helpful to define and explain earlier in the paper.)
Is there a useful way to understand how rapidly the representations move into and out of the bottleneck dimension in the infinite effective depth limit?
Can the authors comment on how the theory might be useful when the true bottleneck rank isn’t known a priori (i.e. some well-used dataset), or when the bias towards bottleneck structure is (mal)-adaptive?

Minor comments:
The legibility (mainly caption and legend size) of the figures could be improved. It might also be helpful to explain the bottleneck structure exemplified in Fig. 2b a bit more, considering it is less visually obvious than for the one presented in Fig. 1b

“One of our goal…” — the first part of this sentence could be improved.

---

### Meta-Review · Area_Chair_Umef · 2025-02-03

**Recommendation:** Accept (Oral)
**Confidence:** 4

**Metareview:**

A clear accept. All reviewers concur that this paper provides an original, rigorous theoretical framework shedding light on feature learning and bottleneck structures in ResNets. The Hamiltonian formulation is a principled perspective on how the introduced “representation geodesics” evolve. The authors complement theory with illuminating synthetic experiments. The paper iis well-written (with a few minor points that may benefit from further exposition) and the rebuttal has been strong and helpful. It will surely spur further work on feature learning.

---

### Decision · Program_Chairs · 2025-02-11

Accept (Oral)